# Weather Factors Associated with Reduced Risk of Dengue Transmission in an Urbanized Tropical City

**DOI:** 10.3390/ijerph19010339

**Published:** 2021-12-29

**Authors:** Hao Gui, Sylvia Gwee, Jiayun Koh, Junxiong Pang

**Affiliations:** 1Saw Swee Hock School of Public Health, National University of Singapore and National University Health System, 12 Science Drive 2, #10-01, Singapore 117549, Singapore; ephgh@nus.edu.sg (H.G.); ephsgxw@nus.edu.sg (S.G.); mskohjiayun@gmail.com (J.K.); 2Centre for Infectious Disease Epidemiology and Research, National University of Singapore, 12 Science Drive 2, #10-01, Singapore 117549, Singapore

**Keywords:** dengue, air quality, wind speed, temperature, rainfall

## Abstract

This study assessed the impact of weather factors, including novel predictors—pollutant standards index (PSI) and wind speed—on dengue incidence in Singapore between 2012 and 2019. Autoregressive integrated moving average (ARIMA) model was fitted to explore the autocorrelation in time series and quasi-Poisson model with a distributed lag non-linear term (DLNM) was set up to assess any non-linear association between climatic factors and dengue incidence. In DLNM, a PSI level of up to 111 was positively associated with dengue incidence; incidence reduced as PSI level increased to 160. A slight rainfall increase of up to 7 mm per week gave rise to higher dengue risk. On the contrary, heavier rainfall was protective against dengue. An increase in mean temperature under around 28.0 °C corresponded with increased dengue cases whereas the association became negative beyond 28.0 °C; the minimum temperature was significantly positively associated with dengue incidence at around 23–25 °C, and the relationship reversed when temperature exceed 27 °C. An overall positive association, albeit insignificant, was observed between maximum temperature and dengue incidence. Wind speed was associated with decreasing relative risk (RR). Beyond prevailing conclusions on temperature, this study observed that extremely poor air quality, high wind speed, minimum temperature ≥27 °C, and rainfall volume beyond 12 mm per week reduced the risk of dengue transmission in an urbanized tropical environment.

## 1. Introduction

Dengue is a mosquito-borne viral disease that is endemic in tropical and subtropical regions. Globalization and global warming, however, have facilitated its expansion to non-endemic countries over the years [1,2]. An estimated 50 to 200 million dengue infections occur annually, with more than half the world at risk of infection [3]. The disease may escalate and cause dengue hemorrhagic fever (DHF) in a small minority, and the annual death toll from dengue lies between 20,000 and 25,000. Four antigenically distinct serotypes (DENV1/2/3/4) co-circulate globally, and DHF is more often associated with secondary infections [4].

The underlying causes of dengue epidemics in endemic countries are multifactorial [5] and are reflective of the interplay between host, vector, virus, and environment in disease transmission. Factors that drive epidemics include a change in predominant serotype [6,7], or lowered herd immunity due to waning immunity, and increased proportion of susceptible individuals accompanying population replacement [8,9]. Climatic factors are also integral to dengue transmission, as reflected by the seasonal nature of the disease [10,11].

*Aedes aegypti* (*Ae. aegypti*), the more efficient vector of dengue virus than *Aedes albopictus*, is adapted to peri-domestic environments [12,13]. In urban settings, outdoors breeding of *Ae. aegypti* in rainwater accumulating containers gives an indication of how climatic factors support environment factors that influence vector growth [14,15]. There is much evidence that meteorological factors including temperature, rainfall, humidity, and air quality influence vector growth and distribution both directly and indirectly. Studies have found that a temperature rise of up to 34 °C increases all stages of *Ae. aegypti* developmental rates, resulting in population growth [16]. Increased dengue transmission under warmer temperatures can arise from faster viral replication within the vector, shortened extrinsic incubation period, and increased feeding rate of *Ae. aegypti* [16,17,18]. Humidity also increases viral propagation and hatch percentage of *Ae. aegypti* eggs [19,20]. Human practices coupled with climatic factors, such as water storage in dry climates, can encourage mosquito productivity, and dengue transmission [21]. This is exemplified by Schmidt et al.’s findings of higher dengue risk in rural areas characterized by the absence of tap water supply [22].

In general, the significant effect of warmer temperatures on increased dengue rates is largely consistent across studies, while the role of humidity, air quality, rainfall, and haze on dengue transmission is not clear and less often studied [6,23,24]. A comprehensive understanding of how meteorological factors influence vectors and humans is crucial for disease forecasting and control. Additionally, it has been recognized that predictive models for dengue transmission need to account for the interaction between climatic, host-vector specific, and viral factors.

Singapore is a tropical country where dengue resurgence typically occurs in a 5–6-year cycle, but the last decade has seen dengue escalate to record numbers [7,25]. Despite aggressive vector control programs and public awareness campaigns, dengue cases surpassed the tens of thousands between 2013 and 2019 [7,26,27]. Another strategy adopted by the Singapore government has been to publicize data on the real-time distribution of cases and hotspots as part of timely alerts to the community [7,25,28]. Nonetheless, evidence has shown that vector control is the most effective means to reduce dengue transmission [9], and it is envisaged that predictive surveillance and the targeted response would be an efficient long-term strategy.

Hence, this study aims to assess the association between multiple weather factors and dengue incidence in Singapore between 2012 and 2019, as well as examine the roles of less well-characterized weather variables including air quality and wind speed. This would help inform dengue predictive models in the region and guide targeted disease control efforts.

## 2. Materials and Methods

### 2.1. Data Collection

Weekly notified dengue cases in Singapore from 1 January 2012 to 25 August 2019 were collected from the Weekly Infectious Diseases Bulletin published by the Ministry of Health [28]. Daily weather data collected include the mean, maximum and minimum temperature (°C), total rainfall (mm), mean wind speed (km/h), and pollutant standard index (PSI), all of which were obtained from open sources released by the National Environment Agency [29]. The data were collected across meteorological stations evenly distributed over Singapore. City-wide data points were aggregated by averaging across all stations while all daily figures were averaged across each reporting week to obtain weekly-representative data such that it was on a consistent scale with the dengue cases provided by the Ministry of Health. The total population data were based on the mid-year human population of the respective year from the Singapore Department of Statistics [30].

### 2.2. Statistical Analysis

#### 2.2.1. Autoregressive Integrated Moving Average (ARIMA) Model

Autoregressive integrated moving average (ARIMA) is a class of models that “explains” a given time series based on historical values. For initial analysis, the univariate ARIMA model was built to capture the time structure of dengue cases. Weather variables with appropriate lags were added as exogenous variables to improve the model. The equation used for ARMA (p, q) model is as follows:(1)$$y_{t}=c+a_{1}y_{t-1}+\ldots +a_{p}y_{t-p}+u_{t}+m_{1}u_{t-1}+\ldots +m_{q}u_{t-q}$$
where $y_{t}$ is the target time series value at time *t*; *c* is constant; $u_{t},\;u_{t-1},\;\ldots ,\;u_{t-q}$ are white noise error terms; $a_{i}$ (*i* = 1, 2… *p*) and $m_{i}\;$ (*i* = 1, 2… *q*) are corresponding parameters.

Since the time-series graph (Figure 1) indicated a clear long-term trend and non-stationary trait, first-differencing of the data was conducted to build the ARIMA model. Weekly dengue cases numbers were then subjected to natural log transformation to stabilize the variance. The seasonal ARIMA model was also investigated but results were placed in the supplementary due to drawback of weekly data—the seasonal period is not a common number usually representing one year (52 weeks).

The cross-correlation function (CCF) graphs showcase all-weather factor correlations with weekly dengue incidence while accounting for lagged effects, at a significance level of 0.05. Due to the autocorrelation of both time series, inference of coefficient’s correlation may not be robust and pre-whitening technique was applied to circumvent the issue [31]. Pearson correlations were considered for a linear relationship. Of all the models tested, an ARIMA(1, 1, 0) model was found to best fit our data. The equation ARIMA(1, 1, 0) with exogenous variable is as follows:(2)$$y_{t}^{\prime }=\log\left(y_{t}\right),\;y_{t}^{\prime }=u_{t}+\alpha_{2}x_{t-l},\;\nabla u_{t}=\alpha_{1}\nabla u_{t-1}+w_{t}$$
where $\alpha_{1},\;\alpha_{2}$ are coefficients of the autoregressive term and exogenous variables, respectively; $x_{t-l}$ refers to exogenous weather values with l weeks lag effect; $u_{t}$ refers to the residual of regression; $\;\nabla u_{t}$ is the residual with one-order differencing; $w_{t}$ is the white noise error term. Only univariate weather factor models were constructed here.

#### 2.2.2. Distributed Lag Non-Linear Model

Recognizing that relationships between weather and dengue are not simply linear in practice, non-linear associations were also explored using a distributed lag non-linear model (DLNM) [32]. DLNM is an advanced statistical method developed to simultaneously estimate the nonlinear and delayed effects of the concerning exposure on response. The cross-basis function describes a two-dimensional relationship along the dimensions of factor and distributed lag is the core of DLNM implementation. Compared to other methods, DLNM has three advantages [33]. First, beyond the usual exposure-response relationship, exposure-lag-response defined by Antonio Gasparrini introduced additional temporal dimensions needed to express the association. Second, nonlinear dependencies capture more complicated patterns than linear exposure–response relationships. Third, DLNM has been evaluated and confirmed to offer a well-grounded performance regarding complex estimation routines, as well as a comprehensive scheme for interpretation.

The number of weekly dengue cases was assumed to follow an over-dispersed Poisson distribution [34]. Thus, quasi-Poisson regression coupled with DLNM cross-basis was considered. Univariate exposure analysis was conducted first for simplicity. The model equation in this study is [19] as follows:(3)$$\log\left(E\left(y_{t}\right)\right)=\beta_{0}+\beta_{1}\log\left(y_{t-1}\right)+\beta_{2}\log\left(y_{t-2}\right)+s_{1}\left(x_{t,\;j},\;l_{j},\;\delta_{1},\;\delta_{2},\Phi_{j}\right)+s_{2}\left(t,\;\delta ,\Psi \right)+\log\left(N_{t}\right)$$

$y_{t}$ is the number of dengue cases at week t, while $\beta_{1}$ and $\beta_{2}$ are the coefficients of the autoregression terms—two autocorrelation terms were introduced into models as the ARIMA model and residual analysis indicated that dengue cases from the past two weeks would be significantly correlated with a current number of cases. $s_{1}\left(x_{t,\;j},\;l_{j},\;\delta_{1},\;\delta_{2},\Phi_{j}\right)$ is the cross-basis of distributed lag non-linear terms with $x_{t,\;j}$ representing a single weather factor at week *t*, and $l_{j}$ representing the maximum lag number. A natural cubic spline (ns) smoothing method with $\delta_{1},\;\delta_{2}$ degree of freedom (df) was used to describe the non-linear effect in both weather factor and lag time, respectively, with $\Phi_{j}$ denoting corresponding coefficients. $s_{2}\left(t,\;\delta ,\Psi \right)$ is another natural cubic spline term to describe the long-term trend and seasonality of dengue cases time series, with δ,Ψ representing degree freedom and corresponding coefficients. Here, the degree freedom is assigned to be 1 per year, therefore the δ equals to duration of the study period in years (approximately 8 years). $\log\left(N_{t}\right)$ is the offset term that considers the effect of differences in Singapore’s mid-year population. Notation “log” in Equation (2) represents the natural log transformation.

Quasi Akaike’s information criterion (*QAIC*) was applied to evaluate the goodness of fit.
(4)$$QAIC=-2ℒ\left(\hat{\Theta }\right)+2\hat{\phi }k$$

As shown in the equation above, $ℒ\left(\hat{\Theta }\right)$ is the log-likelihood of estimated parameters, $\hat{\phi }$ is the estimated over-dispersion parameter and k is the number of parameters. Instead of using the same degree of freedom for all-weather factors, we applied the grid search method to derive the best degree of freedom for each factor, which satisfied both parsimony and fit accuracy based on *QAIC*. Thereafter, values between 2–6 were selected for $\delta_{1}$ term in Equation (2), while values between 3–5 were selected for $\delta_{2}$. The selected values for maximum lag numbers $l_{j}$ in Equation (2) was 0–16 weeks (i.e., roughly four months) [35,36].

Relative risk (RR) with 95% confidence intervals (CI) was selected to be the measure of effect. The exposure–response relationship estimated by DLNM was visualized with graphical representations, including 3D graphs, contour graphs, sliced graphs, and overall effect graphs [32]. 3D and contour graphs display the variation of RR along with 2-dimensional basis exposure and lag. Sliced graphs extract one slice from 3D graphs to emphasize relationships at one specific exposure value or lag. Overall effect graphs were used to measure the overall performance of association by summing up the RR of each lag. The median of each climatic variable was used as the reference, in line with the aim of examining non-linear trends with possible threshold effects.

#### 2.2.3. Sensitivity Analysis

In addition, sensitivity analysis was performed by changing the degree of freedom (df) of natural cubic spline terms to check the robustness of the model. The degree freedom of time (δ) was changed by setting 2 and 3 df per year to interpret the potential seasonality pattern. Moreover, we replaced the natural cubic spline with a polynomial function with different degrees to depict the nonlinear association between the weather factor and dengue incidence. The benchmark of the judgement was based on overall effect graphs.

All statistical analyses were performed on R software (version 3.6.1; R Foundation for Statistical Computing, Vienna, Austria.), and included the use of package “dlnm”, version 2.3.9 [37].

## 3. Results

### 3.1. Epidemiological Characteristics of Epidemics between 2012 and 2019 in Singapore

Between January 2012 and August 2019, there were a total of five major dengue epidemics with the incidence above 100 per 100,000 in Singapore [38]. The time-series data of dengue case trend over the study period illustrates epidemics characterized by peaks in 2013, 2014, 2015, 2016, and 2019 (Figure 1). The average weekly number of dengue infections was 217.02. Generally, there were small variations in weekly temperature and wind speed, whereas weekly rainfall and PSI varied across a wider range (Table 1).

The 2013 epidemic holds the current national record for the highest number of dengue cases in a year, where the dengue incidence was 22,170 cases (404.9 cases per 100,000 population annually) and dengue 1 serotype was the predominant serotype (Appendix A). Thereafter, high incidence levels continued to be seen in the 2014 epidemic, which became Singapore’s second-largest outbreak within a year—the dengue incidence was 18,326 cases (325.6 cases per 100,000 population annually), with dengue 1 serotype as the predominant serotype. The 2015 epidemic was comparatively mild, with dengue incidence at 11,294 cases (196.1 cases per 100,000 population annually) and was predominated by serotype 2. Subsequently, the 2016 epidemic saw a dengue incidence of 13,085 cases (229.1 cases per 100,000 population annually) and was predominated by dengue 2 serotype. After low dengue activity levels for a couple of years, the 2019 epidemic became the third-biggest dengue outbreak recorded nationally with a dengue incidence of 16,100 cases (282.3 cases per 100,000 population annually), and dengue 2 and 3 serotypes taking over as the co-predominant serotypes [30,39,40].

### 3.2. Linear Relationship with Weather Factors in Cross-Correlation and ARIMA Modelling

Pearson’s cross-correlation coefficients for each climatic factor after pre-whitening are shown in Table 2 and cross-correlation graphs can be seen in Appendix A. In general, pollutant standard index (PSI) five weeks and seven weeks before the dengue onset was significantly and negatively correlated with dengue cases. In addition, the mean wind speed was found to have a significant negative correlation with five-week delayed dengue cases. As for temperature indicators, mean temperature (lag effect after 1, 2, 11 weeks), maximum temperature (lag effect after 11, 13, 14 weeks), and minimum temperature (lag effect after 1, 2, 3 weeks) were positively correlated with weekly dengue cases. No significant linear correlation was observed between rainfall and dengue incidence in Singapore.

Since a significant autocorrelation coefficient was found at a lag time of 1 week as indicated by the autocorrelation graph (Appendix A), ARIMA (1, 1, 0) with first-order differencing and one autoregressive term was selected as the best model to describe the trend and autoregressive parameters of the dengue time series. Residual diagnostics were conducted to assess model appropriateness, including the use of Ljung-Box tests (*p*-value = 0.3686) (Appendix A). The Akaïke information criterion (AIC) of ARIMA(1, 1, 0) was −128.47.

Using univariate ARIMA models with covariates derived from significant cross-correlations, the following factors were found to be significantly associated with dengue incidence: 5-week and 7-week time lag effect of PSI, 1-week and 11-week time lag effect of mean temperature, an 11-week time lag effect of maximum temperature and a 5-week time lag effect of mean wind speed (Table 2). Among the above significant covariates, PSI and mean wind speed showed protective effects from dengue cases, while others displayed positive effects. All models with statistically significant exogenous variables had lower AIC than simple ARIMA(1, 1, 0) with no exogenous variable, which indicates an improvement in the fitting. The insignificance of covariates suggested by cross-correlation might be attributed to a 5% confidence level, which implies a 5% possibility of wrong signals.

### 3.3. Non-Linear Relationship with Weather Factors in Cross-Correlation and DLNM

For the quasi-Poisson model with distributed lag non-linear cross-basis terms, parameter settings with optimal fitting performance (the smallest *QAIC*) were chosen. The choice of degree of freedom, parameters, and corresponding *QAIC* of each weather factor are shown in Appendix A. The models in the univariate analysis included the effect on dengue cases by allowing for PSI (lag 0–16 weeks), total rainfall (lag 0–15 weeks), mean temperature (lag 0–16 weeks), maximum temperature (lag 0–16 weeks) and wind speed (lag 0–5 weeks), respectively. Autocorrelation of residuals was tested to ensure the absence of autocorrelation and the assumption of overdispersion was checked (Appendix A), justifying the use of quasi-Poisson regression. The fitted number of dengue cases predicted by our models mirrored the observed trend of dengue cases closely as seen in Figure 2. The graphical output of dengue DLNM modeling for each climatic variable is described in the following subsections. Additionally, we also conducted predictive models using 2012–2018 data as trial data, and 2019 data as test data. Details of the predictive models and their performance are detailed in the Appendix A).

#### 3.3.1. Effect of PSI

With a weekly PSI median of 48.22 as a reference, visualization of DLNM model output with the 3D graph and contour graph (Figure 3a,b) showed that the RR of dengue was generally higher when PSI (1) reached around 60–140 in the same week (lag0), and (2) was around 60–120 with a lag time of approximately 6–13 weeks. On the contrary, RR of dengue was lower when PSI (1) exceeded approximately 140 regardless of any lag time, and (2) was within 60–140 with a lag time around 2–4 weeks.

The sliced graph for PSI at lag 7 (Figure 3c) indicated that increased PSI between 15 and 106 was associated with increased RR of dengue (RR at PSI of 106: 1.041, 95% CI: 1.016, 1.066), and PSI beyond 106 was associated with dengue reduction (RR at PSI 160: 0.811, 95% CI: 0.725, 0.906). While the lag-response association differed across different PSI levels (Figure 3d), a consistent protective effect was observed across high PSI levels beyond 100 between the lag time of 0–2 weeks.

The overall effects graph (Figure 3e) exhibited a similar trend as the sliced graphs at lag 7. A PSI level of up to 111 was positively associated with dengue incidence (RR at PSI 111: 1.339, 95% CI: 1.020, 1.757), while the subsequent increase in PSI level to 160 was associated with a reduction in dengue incidence (RR at PSI 160: 0.116, 95% CI: 0.031, 0.442).

#### 3.3.2. Effect of Rainfall

The median of weekly total rainfall (5.54 mm) was taken as the reference. The 3D and contour graphs (Figure 4a,b) demonstrated the same fluctuation between positive and negative associations between exposure and response across all lags. Although the sliced graphs at lag 4 and lag 14 had similar curves, we observed slightly different confidence intervals, which suggested that rainfall with a 4-week lag time had a more significant positive effect on dengue incidence when the volume of total rainfall was low whereas rainfall with 14-week lag time had a more significant negative effect when the volume of total rainfall was relatively higher (Figure 4c). The overall effect graph showed a similar trend as with the sliced graphs, with rainfall of 7 mm per week associated with increased dengue incidence (Rainfall: 7 mm, RR: 1.21, 95% CI: 1.06–1.38) (Figure 4d). On the contrary, heavier rainfall had an increasing “protective” effect against dengue, with RR: 0.45, 95% CI: 0.29–0.72 at rainfall level 12 mm per week and RR: 0.17, 95% CI: 0.06–0.50 at rainfall level 22 mm per week. An increased RR, albeit insignificant, was observed at rainfall levels of 15–18 mm, indicating a higher risk of dengue at this interval compared to the reference.

#### 3.3.3. Effect of Temperature

The median of weekly mean, maximum and minimum temperatures were 28.0 °C, 31.8 °C and 25.1 °C, respectively. From the 3D and contour graphs (Figure 5a,b), all three types of temperature showcased similar behavior in exposure-lag-response relationship. Each of them presented a different temperature-dengue correlation at each lag. Lags 1 and 5 were selected as the benchmark to illustrate the variation of association at different lags. The RR increased when mean, maximum and minimum temperatures increased with one-week lag. Such findings correspond with our conclusions from linear ARIMA model whereby there is a general positive association between temperature and incidence at around lag 1. On the other hand, the RR decreased once temperatures reached a higher level than their respective references at lag 5 (Figure 5c–e). Lag-response association, holding the temperature constant, behaved similarly in all three types of temperature (Figure 5c–e). At low temperatures (mean temperature 26 °C, maximum temperature 29 °C, minimum temperature 23 °C), we observed a general positive association between dengue cases and every one-week increase in lag numbers despite of statistical insignificance. Prominent “harvesting” effects were observed when the mean, maximum and minimum temperatures were relatively high (29 °C, 33 °C, 27 °C)—a period of excess RR (lag 1) is followed by a negative estimate at longer lags (lag 1–4). In terms of overall effect (Figure 5f), increase in mean temperature under around 28.0 °C corresponded with increased dengue cases whereas the association became negative beyond 28.0 °C (results were not statistically significant). Similarly, minimum temperature was significantly positively associated with dengue incidence at a lower temperature range of around 23–25 °C, and the relationship reversed with RR significantly reduced when temperature goes beyond 27 °C (RR: 0.70, 95% CI: 0.56, 0.88). An overall positive association that was not statistically significant was observed between maximum temperature and dengue incidence.

#### 3.3.4. Effect of Wind Speed

The reference value of wind speed was 7.52 km/h. Different associations between wind speed and dengue were observed at marginal lags and middle lags, as displayed by the 3D and contour graphs of wind speed-lag-dengue association (Figure 6a,b). Meanwhile, high and low wind speeds had distinct relationships with RR. When a one-week lag was considered (Figure 6c), RR increased when wind speed ranged around 5–7.52 km/h and remained constant with a further increase in wind speed. When a 5-week lag effect was factored in, higher wind speed clearly reduced the RR of dengue. A higher wind speed at 9 km/h exhibited a negative effect on RR along with the lags compared to 6 km/h. Overall, the wind speed was associated with decreasing RR (Figure 6d), with a wind speed of 13 km/h associated with an overall reduction in risk of dengue (RR: 0.70, 95% CI: 0.57, 0.87) compared to the reference.

#### 3.3.5. Sensitivity Analysis

In general, varying the degree of freedom (df) of time trend and seasonality did not change the direction of effect to a large degree (Appendix A). However, some statistical significance was lost when df per year increased, such as PSI around 100. The general features of overall effect in Appendix A remained the same when polynomial functions were applied.

## 4. Discussion

This study explored the association between dengue incidence and several weather factors in Singapore between 2012 and 2019, furthering the analysis by Xu and co-authors, who explored whether factors’ impact on dengue in Singapore between 2001 and 2009 [19]. The relationship between weather factors and dengue cases may vary across different periods, especially with increasing global warming and urbanization. Therefore, independent research on these relationships is critical to guide environmental-related policy. Apart from weather variables included in Xu et al., this study considered PSI as a new indicator that could influence the dengue trend. Hussain-Alkhateeb et al. wrote on the successful deployment of an early warning and response system utilizing mean temperature, rainfall, and humidity in the prediction of dengue outbreaks in Brazil, Mexico, and Malaysia; sensitivity and predictive values ranged from 81–99% and 50–88%, respectively [41]. However, their study lacked the exploration of wind speed and PSI level. During the process of model fitting, the degree of freedom parameters and the largest number of weeks were estimated independently in each weather-related model instead of using the same settings of parameters in all models.

In this study, the high temperature was found to be negatively associated with dengue transmission. The reference (median) was reflective of the relatively uniform temperatures that Singapore experiences year-round and increases in temperatures may hinder transmission. However, the risk of dengue increases steeply up to 29 °C and decreases when the temperature is higher than 29 °C. Hence, our study’s results seem to support the idea of an optimum temperature range that promotes dengue transmission, and temperatures outside this range prevent dengue transmission. While we studied the influence of various temperature types on dengue incidence, it is important to note that minimum temperature has proven to be more apt than the other temperature measures due to its consistent significant association with dengue RR (Figure 5). On the contrary, a systematic review of temperature’s effect on dengue risk found that higher mean, maximum and minimum temperatures were individually associated with increased dengue transmission [42]. Another study looking at dengue patterns in Singapore from 1974 until 2011 found that the higher minimum and mean temperature was related to higher dengue transmission, while maximum temperature had no statistically significant relation [43]. On the other hand, Xu et al. reported that dengue incidence in Singapore was amplified by mean temperatures above 27.8 °C (reference value) in 2001–2009, while mean temperatures deviating from 27.8 °C in 2004–2006 and 2007–2009 were associated with reduced dengue transmission [19]. Together with our findings, this corresponds to Connor et al.’s conclusion that *Ae. aegypti* is the most active at 28 °C [44].

Concurrently, molecular studies showed that *Ae. aegypti* survival and development thrives beyond the 30 °C range [45,46]. Others concluded that the ideal range of survival (88–93%) through all phases of development occurs between 20–30 °C [45], including Yang, et al. (2009) who reported 29.2 °C as the best temperature to produce the most offspring [24]. While less is known for Ae. albopictus comparatively, it has a wider developmental temperature range of 15–35 °C; with its shortest gonotrophic cycle occurring at 30 °C; an optimum developmental zero temperature of 29.7 °C; and can survive longer at lower temperatures [44]. One possible explanation for discrepancies between studies could be the differing effects of temperature on various development stages of mosquitos and the dengue virus, accompanied by the dominant strategy of vector control employed at any specific period.

The overall increased effect of temperature on dengue was captured at a lag of one week while elevated temperatures curtailed dengue cases at a lag of five weeks. The five-week lag effect of temperature could be induced by ecological factors pertaining to the Aedes vector such as the length of the gonotrophic cycle, larval development, and growth rate of Aedes mosquitoes. On the other hand, the one-week lag effect of temperature may be a result of behavioral tendencies of Aedes mosquitoes and humans that have repercussions in the shorter term, such as time to blood-feeding or health-seeking behavior after the presentation of symptoms. These associations at specific lag should be emphasized to facilitate warning measures.

The decreased risk of dengue transmission with high wind speeds found in this study was consistent with the results of another study in subtropical city in Guangzhou, China, which reported that wind velocity was inversely associated with dengue incidence in the same month [47]. One potential mechanism that could account for the influence of wind speed is the suppression of mosquitoes’ host-seeking flight activity [48], which reduces oviposition and contact with hosts. As suggested by Hoffman et al. (2002), wind may deter plume following in mosquitoes due to their inability to progress upwind or dilute chemical attractants emitted by the host [49]. Since wind speed affects the mobility of adult mosquitoes, the minimal two-week lag between wind speed and dengue RR observed in this study was consistent with the *Ae. aegypti*’s predicted infective life of 18.26 days at a higher temperature of 30.6 °C by Goindin, et al. [36]. The lag was also aligned with the sum of mean extrinsic and intrinsic incubation periods, which represents the period from the infection of a mosquito to the presentation of symptoms in an infected human, estimated at 6.5 (at 30 °C) and 5.9 days, respectively [50]. Alternatively, wind also directly affects the evaporation rate of both outdoor and indoor vector breeding sites [51], reducing the availability of breeding sites and larval productivity.

Our findings indicate that moderate PSI level (defined as 50–100 by NEA) would increase risk of dengue while unhealthy PSI level (defined as 101–200 by NEA) would inhibit dengue. Extremely high PSI values (>150) reduced the number of dengue cases. This was also supported by a study from Malaysia, where a moderate negative correlation was detected between trapped larvae counts and Air Pollution Index (API) with a one-week lag [52]. Another study conducted in Brazil found particulate matter 10 microns or less (PM10) to have a statistically significant negative correlation with dengue cases [53]. However, a lack of association between dengue activity and haze between 2001 and 2008 were reported from Singapore [24]. It should be noted that the study only utilized the ARIMA model, which explored linear association, whereas this analysis was supplemented with the use of a non-linear quasi-Poisson model. While there is a limited value of this potential association at a policy level to control dengue, this finding still has academic value in strengthening our understanding and prediction of dengue incidence. Smoke, a component of haze, was anecdotally claimed to repel insects from biting [54,55]. The toxicity of haze was believed to reduce mosquito density, as deleterious effect from direct and indirect exposure to haze was found on the development and survival of butterflies [56]. Nonetheless, research pertaining to haze remains sparse. More evidence is required on the mechanisms by which air quality may affect mosquito development or biting habits.

The volume of rainfall has long been viewed as a plausible factor affecting dengue in the domain of public health. Although we spotted some significant intervals showing positive and negative effects alternately across the lags, the conclusion was challenged by variation of CI in a sensitivity analysis. Another study looking at dengue incidence in Singapore from 2000 to 2007 [57] also demonstrated an absence of a relationship between rainfall and dengue incidence [28,29]. One possible explanation could be the role of indoor breeding in urban settings, which is sheltered from outdoor elements. The importance of stagnant water as breeding sites is emphasized by anti-dengue campaigns in Singapore compelling citizens to routinely empty water from flower pots/trays, containers, and closed perimeter drains. According to MOH’s report on vector surveillance in 2018 [38], the top breeding habitats for *Ae. aegypti* included domestic containers (32%), flowerpots/trays (11.1%), ornamental containers (9.3%) and closed perimeter drains (3.7%). On the other hand, a study looking at dengue incidence in Malaysia from 2008–2010 showed that increased bi-weekly accumulated rainfall had a positively strong effect on dengue cases [48]. Yet, experimental and observational studies have shown that excessive rainfall can “flush out” breeding sites and destroy developing larvae, which is consistent with our conclusion. At least two other studies in Singapore have also demonstrated that dengue incidence between 2000 and 2016 was significantly reduced following months where flushing events from wet weather were most frequent [58,59]. These collectively indicate that a more detailed analysis with the use of flood-prone spatial data and its contribution of indoor mosquito breeding, is required to understand the complex role of rainfall patterns on dengue transmission.

In the sensitivity analysis, the reduction in significance was likely due to the competence of more sophisticated spline function for time trend and seasonality, in another words, the higher collinearity between terms in model result in wider confidence interval of relative risk estimate [59]. From the sensitivity analysis conducted, we note with caution that varying parameters of the climatic variables to marginal values that are not commonly recorded may reduce the statistical significance of RR predicted by the model.

### Limitations

Although some long-term significant lags indicated a weak seasonal pattern, the seasonal period and its potential influence on these climatic factors were not studied further. The generalizability of the model may not be applicable to all tropical countries since the model is contextualized to Singapore’s setting. Only short-term predictions can be performed with these results since long-term predictions would be inaccurate and cause larger prediction errors. With climatic variables derived from our quasi-Poisson model, dengue predictions made a week in advance would have moderate performance, as assessed by split-sample model validation. Additionally, this analysis did not include relative humidity and absolute humidity due to the absence of government API data before 2016. In particular, Xu et al. (2014)’s Singapore study concluded that absolute humidity was the most useful weather factor in dengue forecasting.

Moreover, our model did not consider other major factors including circulating dengue serotypes and geospatial differences. In Singapore, the variations of urban housing across its geography, from landed estates to high-rise buildings have been found to influence differences in the dengue distribution [12]. Additionally, weekly meteorological indicators in this analysis were calculated by averaging data across all 63 stations nationwide and do not account for potential geographical differences in weather. Nevertheless, the variation is likely small given the small size of Singapore. Lastly, the study assumes the effectiveness of vector control is constant, regardless of its intensity level, across this period.

## 5. Conclusions

The reduction in risk of dengue transmission was associated with poor air quality, high wind speeds, minimum temperature beyond 27 °C, and excessive rainfall above 12 mm per week. Future studies incorporating geospatial data would be important to understand the relationship between these weather factors and dengue risk. This may guide public health and environmental policy for dengue control in countries with tropical urban environments.

## Figures and Tables

**Figure 1 ijerph-19-00339-f001:**
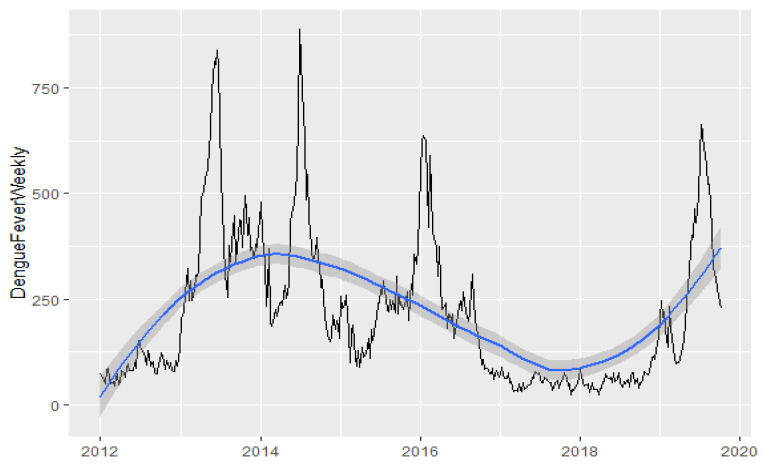
Weekly number of dengue fever cases with smoothed line.

**Figure 2 ijerph-19-00339-f002:**
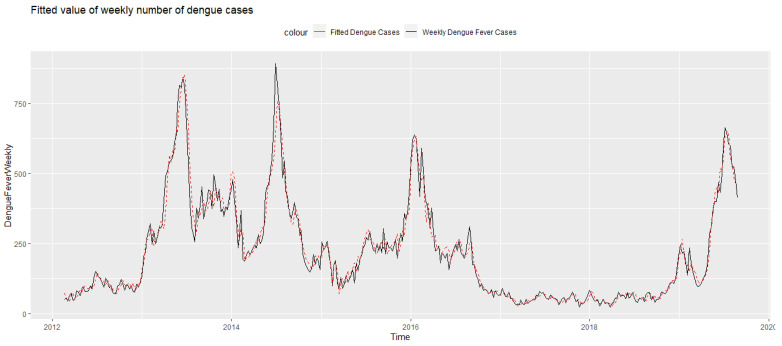
The estimated number of dengue cases (DLNM with wind speed) juxtaposed against the original dengue time series.

**Figure 3 ijerph-19-00339-f003:**
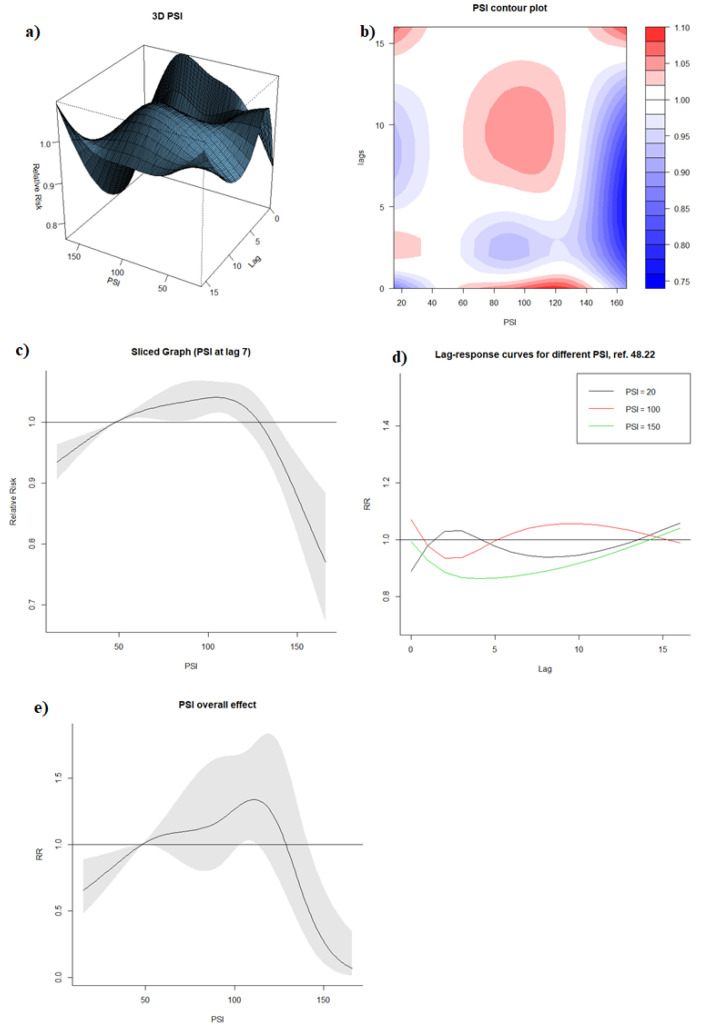
(**a**) 3D graph of PSI-lag-dengue association; (**b**) contour graph of PSI-lag-dengue association; (**c**) sliced graph of PSI-dengue association at lag 7; (**d**) sliced graph of Lag-response association at PSI level of 20, 100, 150; (**e**) overall effect graph of PSI-dengue association.

**Figure 4 ijerph-19-00339-f004:**
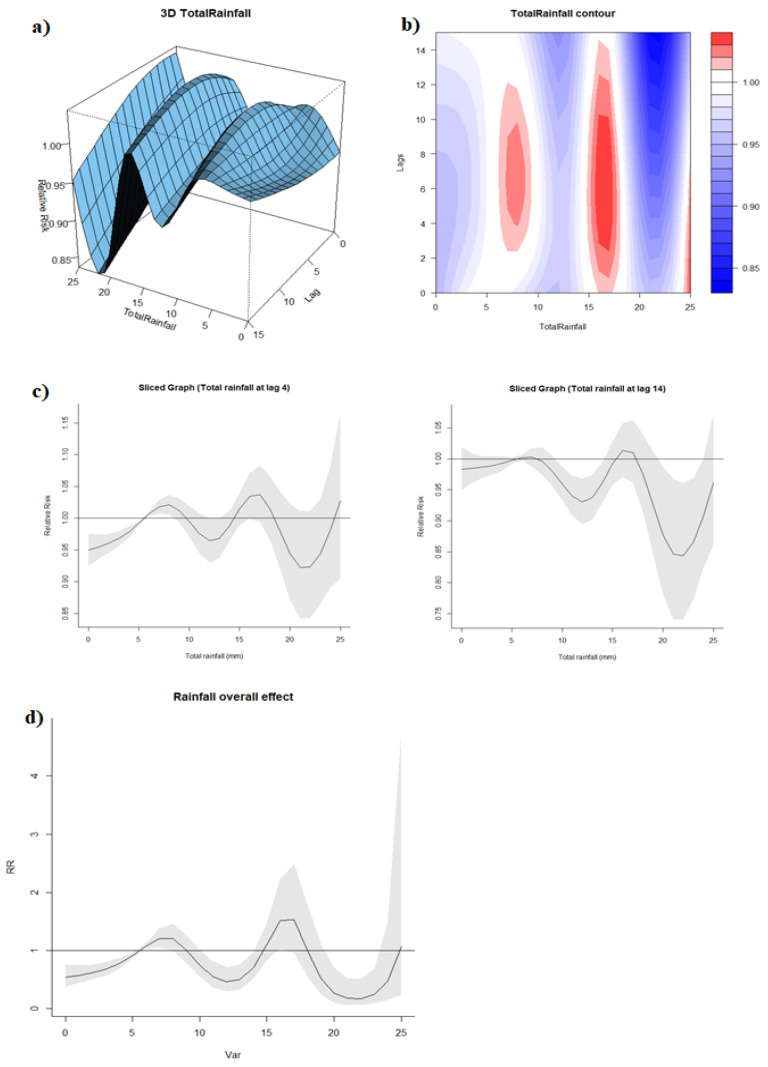
(**a**) 3D graph of rainfall-lag-dengue association; (**b**) contour graph of rainfall-lag-dengue association; (**c**) sliced graph of rainfall-dengue association at lag 4 and 11; (**d**) overall effect graph of rainfall-dengue association.

**Figure 5 ijerph-19-00339-f005:**
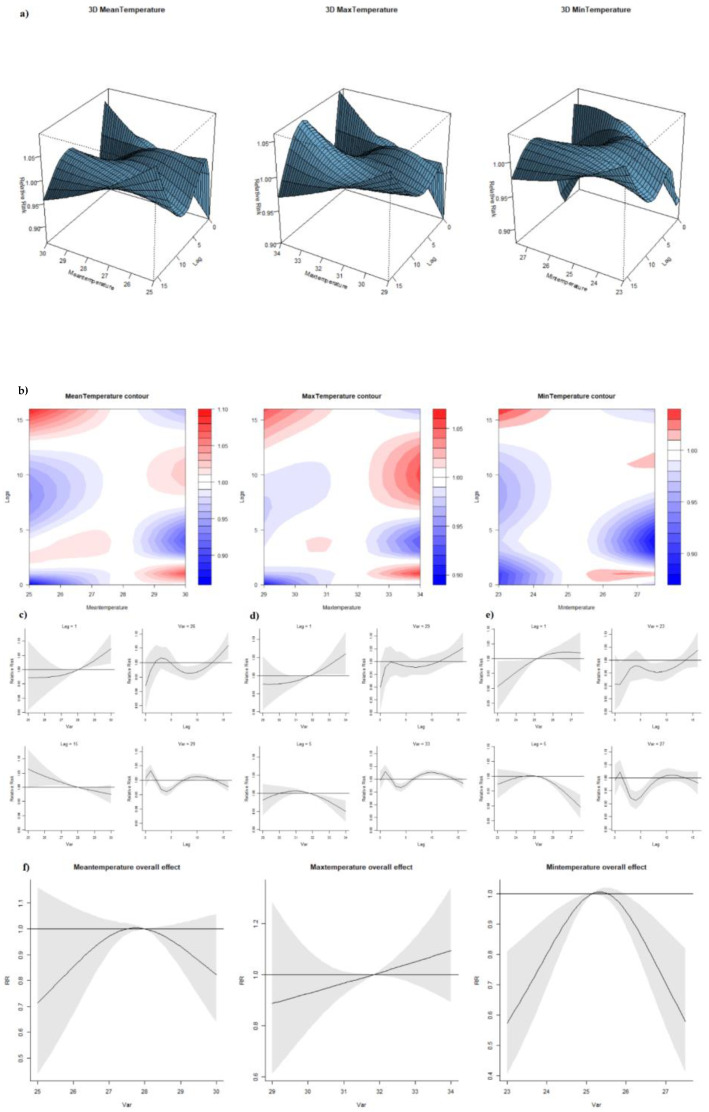
(**a**) 3D graph of temperature-lag-dengue association; (**b**) contour graph of temperature-lag-dengue association; (**c**) sliced graph of mean temperature/lag-dengue association (notes: left top panel shows temperature-dengue association at lag 1 while left bottom one at lag 5; right top panel shows lag-dengue association at mean temperature 26 °C while right bottom one at mean temperature 29 °C; (**d**) sliced graph of max temperature/lag-dengue association (Notes: left top panel shows max temperature-dengue association at lag 1 while left bottom one at lag 5; right top panel shows lag-dengue association at max temperature 29 °C while right bottom one at max temperature 33 °C); (**e**) sliced graph of min temperature/lag-dengue association (Notes: left top panel shows min temperature-dengue association at lag 1 while left bottom one at lag 5; right top panel shows lag-dengue association at min temperature 23 °C while right bottom one at min temperature 27 °C); (**f**) overall effect graph of mean temperature-dengue association.

**Figure 6 ijerph-19-00339-f006:**
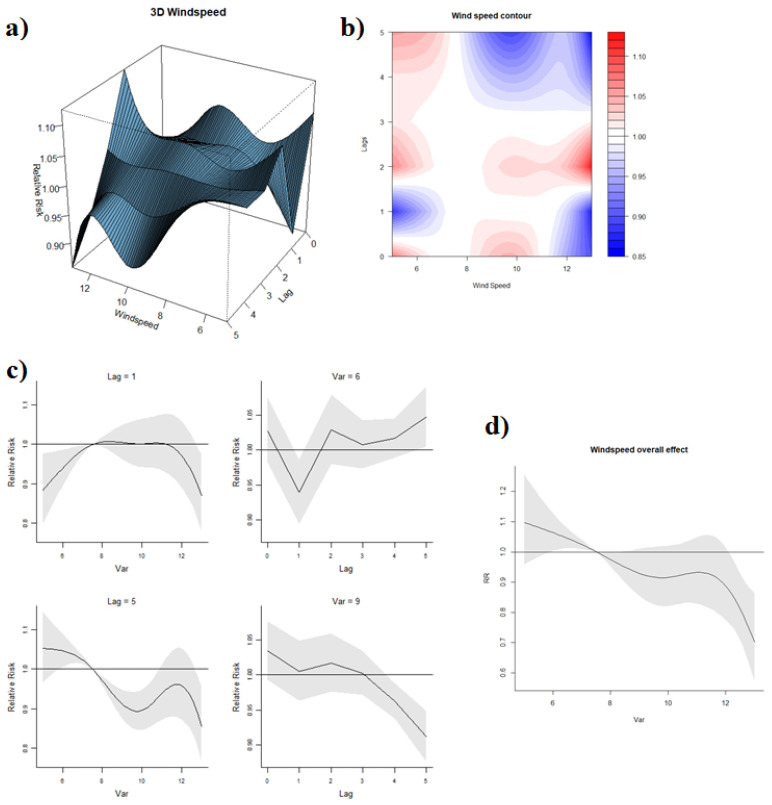
(**a**) 3D graph of wind speed-lag-dengue association; (**b**) contour graph of wind speed-lag-dengue association; (**c**) sliced graph of wind speed/lag-dengue association (Notes: left top panel shows wind speed-dengue association at lag 2 while the left bottom one at lag 5; right top panel shows a lag-dengue association at wind speed 6 km/h while the right bottom one at wind speed 10 km/h); (**d**) overall effect graph of wind speed-dengue association.

**Table 1 ijerph-19-00339-t001:** Descriptive summary of weekly dengue cases and weather conditions in Singapore during 2012–2019.

Variable	Mean	Std	Min	P25	Median	P75	Max	Hist
DengueFeverWeekly	217.02	179.72	24.00	73.75	170.50	303.75	891.00	▇▃▂▁▁
Rainfalltotal	6.37	4.67	0.00	2.72	5.54	8.93	25.36	▇▆▂▁▁
Meantemperature	27.97	0.86	25.04	27.38	28.01	28.51	30.02	▁▃▇▇▂
Maxtemperature	31.79	0.97	27.89	31.25	31.84	32.42	34.19	▁▁▆▇▂
Mintemperature	25.25	0.82	23.17	24.63	25.14	25.79	27.70	▁▇▇▃▁
PSI	46.78	17.15	16.45	35.29	48.22	54.74	165.85	▆▇▁▁▁
Windspeed	7.88	1.74	4.99	6.61	7.52	8.75	14.89	▇▇▃▁▁

P25—25th percentile; P75—75th percentile: PSI—Pollutant Standards Index.

**Table 2 ijerph-19-00339-t002:** Cross-correlation analysis and ARIMA modeling of meteorological factors to dengue incidence.

Meteorological Factor	Lag of Week	Cross Correlation Coefficient	$\mathrm{Autoregressive}\;\mathrm{Coefficient}\;\alpha_{1}\;(p-\mathrm{Value})$	$\mathrm{Coefficients}\;\mathrm{of}\;\mathrm{Exogenous}\;\mathrm{Variable}\;\alpha_{2}\;(p-\mathrm{Value})$	Akaïke Information Criterion (AIC)
PSI	5	−0.111	−0.2128 (<0.001) ***	−0.0020 (0.043) *	−133.68
	7	−0.115	−0.2053 (<0.001) ***	−0.0020 (0.0375) *	−133.91
Mean temperature (°C)	1	0.115	−0.2121 (<0.001) ***	0.0296 (0.0452) *	−131.67
	2	0.100	−0.2175 (<0.001) ***	0.0257 (0.0854)	−130.61
	11	0.118	−0.1970 (<0.001) ***	0.0318 (0.0299) *	−132.37
Maximum temperature (°C)	11	0.155	−0.1995 (<0.001) ***	0.0358 (0.0013) *	−143.4
	13	0.110	−0.2045 (<0.001) ***	0.0031 (0.7838)	−133.1
	14	0.123	−0.2115 (<0.001) ***	0.0164 (0.1421)	−135.19
Minimum temperature (°C)	1	0.126	−0.2024 (<0.001) ***	0.0192 (0.1811)	−126.77
	2	0.114	−0.2064 (<0.001) ***	0.0167 (0.2471)	−126.32
	3	0.103	−0.2085 (<0.001) ***	0.0212 (0.1413)	−127.15
Wind speed (km/h)	5	−0.124	−0.2108 (<0.001) ***	−0.0330 (<0.001) ***	−146.44
Total rainfall (mm)	NS	NS	−	−	

Note: $\alpha_{1}$ is the coefficient of autoregressive term and $\alpha_{2}$ is the coefficient of exogenous weather variables in Equation (2); abbreviations: NS—not significant; * weakly significant; *** highly significant; PSI—Pollutant Standards Index.

## Data Availability

Data presented in this study are available in this article and its Appendix A.

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
