# Peer review of "Weather Factors Associated with Reduced Risk of Dengue Transmission in an Urbanized Tropical City"

_ijerph, 2021, doi:10.3390/ijerph19010339_

Round 1
Reviewer 1 Report
It is a well-written manuscript. I only have a couple of suggestions to improve the manuscript. The inclusion of the pollution standard index(PSI) in the model doesn't fit well with the public health cause; rather, it stands out a bit illogical. Every research has ideological and political implications. So what are we proving here, increased levels of air pollution could be beneficial? The overall message that it hints out is that poor air quality is beneficial in dengue control. At the same time, the evidence is not that strong for such a claim in the present study nor in those you have quoted. The simple availability of data should not be a criterion for inclusion in the model, which could be true in the case of temperature parameters. At least in the discussion session, you should explicitly state the aptness of minimum temperature over other temperature measures. Secondly, I am a bit worried about the potential for overfitting the model, seeing Figure 2. It would be good to do your model with data from 2012 to 2017/18 (trial data) and use the rest to test the model. That would be more appropriate. The minor comments include the need to give full names when using acronyms at the first instance, standard scientific style in quoting species names at all instances, overall proofreading, avoiding discussion points in the results session, etc.
Author Response
Reviewer 1
It is a well-written manuscript. I only have a couple of suggestions to improve the manuscript.
- The inclusion of the pollution standard index (PSI) in the model does not fit well with the public health cause; rather, it stands out a bit illogical. Every research has ideological and political implications. So, what are we proving here, increased levels of air pollution could be beneficial? The overall message that it hints out is that poor air quality is beneficial in dengue control. At the same time, the evidence is not that strong for such a claim in the present study nor in those you have quoted.
Dear reviewer, thank you for your guidance. We agree that this finding has limited policy impact in that it is unethical and impossible to use for vector control but can nonetheless have an academic impact through its use in prediction models. We have since added lines 463-465 in the discussion section to clarify the message we seek to convey. Hope this is ok.
- The simple availability of data should not be a criterion for inclusion in the model, which could be true in the case of temperature parameters. At least in the discussion session, you should explicitly state the aptness of minimum temperature over other temperature measures.
Dear reviewer, thank you for your guidance. We certainly agree that the availability of data does not warrant its inclusion in the model. We would like to clarify that we included multiple types of climatic data seeking to understand the full picture of climate’s influence on dengue incidence, hence we explored all the available variables we could get. We have also clarified this as per your suggestion in lines 397-400 of the discussion section. Hope this suffices.
- Secondly, I am a bit worried about the potential for overfitting the model, seeing Figure 2. It would be good to do your model with data from 2012 to 2017/18 (trial data) and use the rest to test the model. That would be more appropriate.
Dear reviewer, thank you for your suggestion. As for overfitting issue, we think that the fitted line in figure 2 is not that close to the observed dengue cases in that the breaks of y axis is 250 which is a substantial number, in another word, the little deviation in figure 2 may mean larger difference than we thought. We included autoregressive term (lag1, lag2 of dependent variable) as predictors which play a key role in explaining variability of dependent variable. There, we believe that the fitted line does not deviate too much from the observed line. In addition, we applied QAIC (Quasi Akaike s Information Criterion) criterion to select model parameters, which has already considered the penalization on number of parameters to reduce the risk of overfitting.
In this manuscript, we aim to gain a big picture of the association between dengue and weather factors, so we put more emphasis on interpretability of relationship across whole periods of study.
We also conducted predictive model as you said, using 2012 to 2018 as trial data and 2019 data as test data. We have put the details of predictive models as well as their performance (MAE, RMSE) in supplementary material. The general conclusion is that models with weather factors behave better than models without any weather factor in most cases.
- The minor comments include the need to give full names when using acronyms at the first instance, standard scientific style in quoting species names at all instances, overall proofreading, avoiding discussion points in the results session, etc.
Dear reviewer, thank you for your guidance. We have made revisions throughout the manuscript to correct these.

Reviewer 2 Report
In the introduction section - it is better to have description why you chosen mentioned method, what the strength compared with other. In method, It is not clear...whether you used all data in weekly basis or daily basis? an dis it truly comparable?Author Response
Reviewer 2
- In the introduction section - it is better to have description why you chosen mentioned method, what the strength compared with other.
Dear reviewer, thank you for your suggestion. We have elaborated on the rationale behind using the DLNM model, and its strengths in the methods section, lines 134-140. The section was added to the methods instead of the introduction as we believe it would improve the flow better.
- In method, it is not clear...whether you used all data in weekly basis or daily basis? And is it truly comparable?
Dear reviewer, apologies for the confusion. We had aggregated and analyzed all data on a weekly level such that it is compatible with the weekly dengue incidence data provided by the Ministry of Health. The Ministry of Health of Singapore only publishes infectious disease cases on a weekly basis, while the other exploratory variables came in a mix of daily and yearly data. Kindly refer to the figure below for the original data format of all variables. We have also clarified this in the methods sections in lines 89-92. Hope this suffices.

Reviewer 3 Report
My expertise lies in the field of statistics and therefore, my assessment will be concerned with the statistical modelling. I am less able to assess the accuracy of the biological/health implications derived from the results. The main contribution of the paper is to investigate the role of weather variables in the occurrence of dengue cases in Singapore. The procedure of starting with a relatively simple (linear) ARIMA model makes sense. A weak point in the line of argument is the missing comparison between the goodness-of-fit that is possible with the ARIMA approach vs. the quasi-Poisson approach. My detailed remarks are as follows:
Data collection:
All data stem from official sources and seem reliable. Averaging over space is valid, given that the stations are indeed evenly distributed and the considered area is not too large.
ARIMA model:
Figure 1: The (raw) time series is clearly non-stationary. However, the empirical analysis concerns the log-transformed data. Given that the errors from the ARIMA(1,1,0) are (nearly) indistinguishable from white noise (supplement) and that an ARIMA(1,1,0) model allows for a linear, but not a quadratic or cubic trend, I assume that a cubic trend is not significantly present in the log-transformed data. In other words, a plot of the log-transformed series would be more informative.
Table 2:
a_1 and a_2 refer to lag 1 and lag 2 of the dependent variable, according to equation 1. So, it is somewhat misleading that alpha_2 refers to an exogenous variable in Table 2.
It remains a little unclear (to me), which model equations were actually estimated. From my understanding, each row in Table 2 corresponds to a separate model with exactly two parameters: one parameter for the first lag of the dependent variable and one parameter for one lag (not necessarily the first) of only one exogenous variable. If this interpretation is correct, why isn’t there a larger model with possibly multiple lags and multiple variables?
Why don’t the authors employ classical approaches from regression analysis such as the statistical significance of coefficients and information criteria? This would also add consistency to the paper, because information criteria are used for the Quasi-Poisson model. In particular, it would be interesting to know to which extent the addition of those exogenous variables (and possibly their lags) to the ARIMA(1,1,0) leads to an improvement compared to the simple/conventional ARIMA(1,1,0).
Quasi-Poisson model:
From a purely statistical perspective, one may ask why the quasi-Poisson model is needed. They authors should employ some meaningful statistical performance measure to compare it to the (simpler) ARIMA models.
Furthermore, is there any reason for not applying the log-transform here? This step should be beneficial regardless of the model. It would also eliminate the need for a (rather complicated) quasi-Poisson model (because the log-transformed data are not counts anymore)
Finally, I am quite sceptical about the inclusion of the population data. It is only available on a yearly basis, so it cannot be used to explain the week-to-week variability of dengue cases. Furthermore, the population term appears without a coefficient on the right-hand side of the model equation. If population is not an explanatory variable, it has to be moved to the left-hand side (for estimation). It would then imply that the actual dependent variable is the (log of the) number of dengue cases per capita. This would indeed constitute a way to account for a possible population change over time, but this new dependent variable would not be a count variable anymore and the quasi-Poisson model would not be appropriate. So, it needs to be clarified what was actually done.
Weather effects:
I am not familiar with the dlnm framework but the results for the weather effect seem plausible However, the resolution in some of the Figures (such as Figure 5) is too low to be read properly.
Author Response
Reviewer 3
My expertise lies in the field of statistics and therefore, my assessment will be concerned with the statistical modelling. I am less able to assess the accuracy of the biological/health implications derived from the results. The main contribution of the paper is to investigate the role of weather variables in the occurrence of dengue cases in Singapore. The procedure of starting with a relatively simple (linear) ARIMA model makes sense.
- A weak point in the line of argument is the missing comparison between the goodness-of-fit that is possible with the ARIMA approach vs. the quasi-Poisson approach.
Dear reviewer, thank you for your comment. Kindly refer to our detailed response to the weak point argument in the ARIMA model section.
My detailed remarks are as follows:
Data collection:
- All data stem from official sources and seem reliable. Averaging over space is valid, given that the stations are indeed evenly distributed and the considered area is not too large.
Dear reviewer, thank you for your comment.
ARIMA model:
- Figure 1: The (raw) time series is clearly non-stationary. However, the empirical analysis concerns the log-transformed data. Given that the errors from the ARIMA (1,1,0) are (nearly) indistinguishable from white noise (supplement) and that an ARIMA (1,1,0) model allows for a linear, but not a quadratic or cubic trend, I assume that a cubic trend is not significantly present in the log-transformed data. In other words, a plot of the log-transformed series would be more informative.
Dear reviewer, thank you for the comment.
A plot of the log-transformed series as well as first differencing of log-transformed series are provided in supplementary materials figure S2(a)(b). For figure 1, we aim to describe the overview of actual dengue case numbers first and give a big picture for readers. We agree plotting of log-transformed series would be more informative for modelling.
Table 2:
- a_1 and a_2 refer to lag 1 and lag 2 of the dependent variable, according to equation 1. So, it is somewhat misleading that alpha_2 refers to an exogenous variable in Table 2.
Dear reviewer, sorry for the confusion. We have updated the equation in Lines 116-122 with clarification of coefficients.
- It remains a little unclear (to me), which model equations were actually estimated. From my understanding, each row in Table 2 corresponds to a separate model with exactly two parameters: one parameter for the first lag of the dependent variable and one parameter for one lag (not necessarily the first) of only one exogenous variable. If this interpretation is correct, why isn’t there a larger model with possibly multiple lags and multiple variables?
Dear reviewer, your understanding is correct. Here we are concerned about two reasons for not including multiple lags and multiple variables.
- Since weather time series usually have an autoregressive pattern, which means if we add multiple lags terms into one model, multicollinearity issue may arise
- To keep consistent with subsequent DLNM we want to pay more attention to the effect of one single weather factor. Including multiple variables is a plausible way in ARIMA model, but for DLNM model, there are almost larger numbers of coefficients even for just only one weather factor because it includes another dimension – lag. Here we aim to evaluate lagged linear relationship between weather factor and dengue cases.
- Why don’t the authors employ classical approaches from regression analysis such as the statistical significance of coefficients and information criteria? This would also add consistency to the paper, because information criteria are used for the Quasi-Poisson model. In particular, it would be interesting to know to which extent the addition of those exogenous variables (and possibly their lags) to the ARIMA (1,1,0) leads to an improvement compared to the simple/conventional ARIMA (1,1,0).
Dear reviewer, thank you for the great suggestion to include AIC (Akaïke Information Criterion) to compare models. We have made the following additions as per your guidance: in lines 230-231, we added AIC of baseline model ARIMA (1, 1, 0), in lines 238-240, we added comparison between model with exogenous variable and model with no exogenous variable; we also added a column recording AIC of models in table 2.
Quasi-Poisson model:
- From a purely statistical perspective, one may ask why the quasi-Poisson model is needed. They authors should employ some meaningful statistical performance measure to compare it to the (simpler) ARIMA models. Furthermore, is there any reason for not applying the log-transform here? This step should be beneficial regardless of the model. It would also eliminate the need for a (rather complicated) quasi-Poisson model (because the log-transformed data are not counts anymore)
Dear reviewer, thank you for the great comment. First, we used log-transform in ARIMA model for two reasons.
- Stabilize the variance of time series and make it stationary
- Transform count data into a normal distribution to be in line with assumptions of several parametric statistics in ARIMA.
However, ARIMA model is a relatively simple model difficult to catch non-linear association. Absolutely we can add spline terms in ARIMA model, but it is too time consuming to design weather factors with lag effect. Distributed nonlinear lag model embedded in Quasi-Poisson can grasp more association between dengue cases and weather factors.
Why Quasi-poisson model not log-transformed variable using simple regression?
- In tuition, this time series is count data. It is natural to hypothesize count data follows Poisson or negative binomial distribution
- There is one study showing that we should be cautious use of log transformation of count data [1]. Here are some conclusions:
1) Cause bias when handling 0 count using log-transformation.
2) Log-transformed method cause more bias than Poisson, negative binomial model
3) In prediction stage, log transformed method may lead to impossible predictions such as negative numbers of individuals
4) The development of statistical and computational method facilitates fit of Generalized linear model such as Quasi-Poisson model.
According to Chisato Imai et.al [2]. They also mentioned that Gaussian Linear model for log(Yt) has many attractions when counts are consistently reasonably large. For our dataset, dengue cases per week are all larger than 24. So, in this statement we can try simple regression with log(Yt). However, using Poisson regression seems not harmful.
- Finally, I am quite skeptical about the inclusion of the population data. It is only available on a yearly basis, so it cannot be used to explain the week-to-week variability of dengue cases. Furthermore, the population term appears without a coefficient on the right-hand side of the model equation. If population is not an explanatory variable, it has to be moved to the left-hand side (for estimation). It would then imply that the actual dependent variable is the (log of the) number of dengue cases per capita. This would indeed constitute a way to account for a possible population change over time, but this new dependent variable would not be a count variable anymore and the quasi-Poisson model would not be appropriate. So, it needs to be clarified what was actually done.
Dear reviewer, we agree with your observation that the population data cannot be used to explain the week-to-week variability of dengue cases. However, it is difficult to get weekly based population data due to limited source. This log (Population) serves as an offset term in Poisson regression. It constrains the coefficient of this term as 1 and keeps the dependent variable following Poisson distribution. Even if it is synonymous for number of dengue cases per capita(rate), its estimator still derives from generalized Poisson linear regression.
Kindly refer to the following reference [3].
We also observed the use of this offset in model in several studies [4, 5].
Weather effects:
- I am not familiar with the dlnm framework but the results for the weather effect seem plausible However, the resolution in some of the Figures (such as Figure 5) is too low to be read properly.
Dear reviewer, thank you for the comment. We have replaced figures 5 and 6 with clearer resolution ones.
References
[1] O'Hara, R., & Kotze, D. J. (2010). Do not log-transform count data. Methods in Ecology and Evolution, 1, 118-122.
[2] Imai C, Armstrong B, Chalabi Z, Mangtani P, Hashizume M. Time series regression model for infectious disease and weather. Environ Res. 2015 Oct; 142:319-27.
[3] https://towardsdatascience.com/offsetting-the-model-logic-to-implementation-7e333bc25798
[4] Wang Z, Peng J, Liu P, Duan Y, Huang S, Wen Y, Liao Y, Li H, Yan S, Cheng J, Yin P. Association between short-term exposure to air pollution and ischemic stroke onset: a time-stratified case-crossover analysis using a distributed lag nonlinear model in Shenzhen, China. Environ Health. 2020 Jan 2;19(1):1. Doi: 10.1186/s12940-019-0557-4. PMID: 31898503; PMCID: PMC6941275.
[5] Seah A, Aik J, Ng LC, Tam CC. The effects of maximum ambient temperature and heatwaves on dengue infections in the tropical city-state of Singapore - A time series analysis. Sci Total Environ. 2021 Jun 25; 775:145117. Doi: 10.1016/j.scitotenv.2021.145117. Epub 2021 Feb 9. PMID: 33618312.
